# Morphology of Donor and Recipient Nerves Utilised in Nerve Transfers to Restore Upper Limb Function in Cervical Spinal Cord Injury

**DOI:** 10.3390/brainsci6040042

**Published:** 2016-09-27

**Authors:** Aurora Messina, Natasha Van Zyl, Michael Weymouth, Stephen Flood, Andrew Nunn, Catherine Cooper, Jodie Hahn, Mary P. Galea

**Affiliations:** 1Department of Medicine (Royal Melbourne Hospital), The University of Melbourne, Parkville 3010, VIC, Australia; m.galea@unimelb.edu.au; 2Department of Plastic and Reconstructive Surgery, Austin Health, Studley Road, Heidelberg 3084, VIC, Australia; natasha@natashavanzyl.com.au (N.V.Z.); michaelweymouth@me.com (M.W.); steve.flood@live.com (S.F.); 3Victorian Spinal Cord Service, Austin Health, Studley Road, Heidelberg 3084, VIC, Australia; andrew.nunn@monash.edu; 4Department of Occupational Therapy, Austin Health, Studley Road, Heidelberg 3084, VIC, Australia; Catherine.COOPER@austin.org.au (C.C.); Jodie.HAHN@austin.org.au (J.H.)

**Keywords:** peripheral nerves, spinal cord injury, nerve transfer surgery, morphology

## Abstract

Loss of hand function after cervical spinal cord injury (SCI) impacts heavily on independence. Multiple nerve transfer surgery has been applied successfully after cervical SCI to restore critical arm and hand functions, and the outcome depends on nerve integrity. Nerve integrity is assessed indirectly using muscle strength testing and intramuscular electromyography, but these measures cannot show the manifestation that SCI has on the peripheral nerves. We directly assessed the morphology of nerves biopsied at the time of surgery, from three patients within 18 months post injury. Our objective was to document their morphologic features. Donor nerves included teres minor, posterior axillary, brachialis, extensor carpi radialis brevis and supinator. Recipient nerves included triceps, posterior interosseus (PIN) and anterior interosseus nerves (AIN). They were fixed in glutaraldehyde, processed and embedded in Araldite Epon for light microscopy. Eighty percent of nerves showed abnormalities. Most common were myelin thickening and folding, demyelination, inflammation and a reduction of large myelinated axon density. Others were a thickened perineurium, oedematous endoneurium and Renaut bodies. Significantly, very thinly myelinated axons and groups of unmyelinated axons were observed indicating regenerative efforts. Abnormalities exist in both donor and recipient nerves and they differ in appearance and aetiology. The abnormalities observed may be preventable or reversible.

## 1. Introduction

Nerve transfers are increasingly being used to restore volitional control to arm and hand muscles in patients with a cervical spinal cord injury (SCI) [1,2]. To date, up to three nerve transfers have been successfully carried out simultaneously in a single limb, to increase the number of limb functions restored [1]. Common transfers used include the teres minor (TM) to triceps nerve for elbow extension [3], the supinator to posterior interosseous nerve (PIN) for finger and thumb extension [4], the extensor carpi radialis brevis (ECRB) to flexor pollicis longus nerve [5], and the brachialis nerve branch to the anterior interosseous nerve (AIN) for extrinsic thumb flexion and reanimation of flexor digitorum profundus to the index and middle finger [6,7].

Selection of candidates for nerve transfer surgery is based on the availability of supralesional donor nerves that are uninjured, redundant and have intact upper motor neuron (UMN) and lower motor neuron (LMN) pools; and of infralesional recipient nerves that are intact and innervate muscles that are responsive to electrical stimulation [1]. The overall integrity of nerves and muscles is determined indirectly using muscle strength evaluation according to the Medical Research Council grading [8], and intramuscular electromyography [1,9]. During surgery, connectivity of donor nerves to the motor cortex is ascertained by measuring action potentials following transcranial electrical stimulation, and recipient nerve/muscle function is assessed by visualisation of muscle contraction and nerve conduction measurement following direct stimulation [7].

It is widely assumed that the spinal motor neuron pools and their corresponding peripheral nerves outside the zone of SCI are normal. However, the zones of spinal cord injury are not clearly demarcated nor are they symmetrical, and, hence, peripheral nerve involvement is not readily predicted [10]. Suboptimal electromyography recordings in the form of reduced compound muscle action potentials (CMAP), muscle fibrillations and altered nerve conductance such as reduced velocity and amplitude have been widely reported in SCI [11,12]. Their aetiology is not understood when based on the assumption that spinal motor neurons and peripheral nerves are normal. To date, direct assessment of peripheral nerve morphology has been hampered by the lack of interest in cadaver studies and the lack of peripheral nerve availability [13]. AIN morphology, post SCI, has been described in one study series where a decreased fibre density in 2/7 nerves was reported [14].

As the number of nerve transfers and other SCI therapies are on the increase, we consider it critical to directly evaluate the extent and manifestation of SCI with respect to the peripheral nerves, since success for any technique that effectively bypasses the injury in the cord depends on their health. Nerve transfer procedures present a novel, unique and rare opportunity to sample multiple peripheral nerves without patient morbidity. Direct morphologic assessment is essential to increase our understanding about the impact that SCI has on the peripheral nervous system, guide us towards interventions that may assist in maintaining nerve health, and shed light on the aetiology of observations made using indirect assessments.

## 2. Patients

Ethics approval, number HREC/13/Austin/245, was obtained from the Austin Health Human Research Ethics Committee. Research was conducted in accordance with the National Statement on Ethical Conduct in Human Research 2007 (Updated May 2015) and the World Medical Association Declaration of Helsinki. Informed consent for recruitment to the Nerve Transfer Study was obtained from each patient.

## 3. Materials and Methods

Clinical trial number: ACTRN12615000179538. A total of twenty donor and recipient nerve specimens (1–2 mm long) were collected upon surgery from three participants who had sustained cervical spinal cord injuries and were referred for surgical reconstruction of hand function. Multiple transfers were carried out using: supinator nerve to PIN [4], brachialis or ECRB to AIN, [7] and teres minor (TM) and/or posterior axillary (Pax) nerve to triceps [15]. Specimens were gently stripped of any surface fat and immediately immersed in 2.5% glutaraldehyde in 0.1 M phosphate buffer for 48 h, washed, post fixed in 1% osmium tetroxide solution and washed again. They were dehydrated in increasing concentrations of acetone then impregnated and embedded in Araldite Epon according to standard protocols. Semi-thin 0.8 to1 μm thick sections were cut using an ultramicrotome (AF Huxley, Cambridge, UK), stained with 1% Methylene Blue or 1% Toluidine Blue and cover-slipped for light microscopy.

## 4. Results

Multiple sections (10–20) from each sample were assessed by light microscopy at ×40 to ×1000 magnification, using an Olympus BX60 microscope (OLympus, Denmark) and images were captured using an Olympus DP71 camera. Qualitative data were recorded from the 20 nerve specimens, according to various morphologic criteria. A semi-quantitative assessment was carried out to in order to estimate the proportion of fibres exhibiting myelin abnormalities. The total endoneurial area of one section from each patient was assessed. A 10 × 10 (1 cm square) grid was placed into the microscope eyepiece. Specimens were viewed at ×100 or ×200 magnification depending on their size. The number of points associated with abnormal and normal myelin was counted and recorded separately. The number of points associated with abnormal myelin was divided by the total number of points counted, and expressed as a percentage.

Patient demographics are described in Table 1 and the qualitative data for nerves are summarized in Table 2.

Table 1 shows the age, gender, time post injury (TPI) and American Spinal Injury Association (ASIA) score for neurologic assessment for patients with a spinal injury.

Table 2 shows a summary of the nerves assessed. Abnormalities were observed in both donor and recipient nerves. In recipient nerves, we observed a high incidence of myelin abnormalities and a reduction in the density of large myelinated fibres. In the donor nerves, we observed a high incidence of reduced axon density, the generation of Renaut bodies but few myelin abnormalities. Both the recipient and donor nerves exhibited mononuclear cell infiltration with the predominant cell being a monocyte (Table 2).

## 5. Donor Nerves

### 5.1. Supinator (n = 4)

Supinator nerves were comprised of up to four neatly demarcated circular fascicles. In all specimens, populations of large myelinated, small thinly myelinated and unmyelinated axons were observed (Figure 1c,d,f,g). The morphologic appearances of the first and second specimens were relatively normal. They showed a large myelinated axon population with a uniform orientation, axon density, myelin thickness and perineurium thickness (Figure 1a). In the third specimen, myelination was markedly abnormal in one of the three fascicles and showed mild changes in another (Figure 1b,c,d). Axons were surrounded by a thickened or distorted myelin sheath. In many fibres, the myelin was also “folded,” giving rise to superfluous myelin that jutted inwards encroaching heavily on the axon space. In other fibres, the dark staining myelin was separated or “split” longitudinally into two sections by a pale stained region that often extended around most of the circumference. (Figure 1c,d). Some perineural vessels were filled with mononuclear cells mostly comprised of monocytes. (see Figure 1h). In the fourth specimen, the large axon density was markedly reduced (Figure 1e,f), the endoneurium was sparsely populated, and the sub-perineural tissue appeared oedematous in some regions. Many of the large myelinated axons appeared relatively normal but thickened, folded and thinned myelin abnormalities were noted in others (Figure 1f,g). Mononuclear cells were present in blood vessels in the perineural and sub-perineural regions of abnormal nerves (Figure 1h).

### 5.2. Brachialis (n = 1) and ECRB (n = 1)

The brachialis nerve appeared relatively normal (Table 2), and the ECRB nerve fibre morphology appeared relatively normal, although the large myelinated axon density was notably reduced. Some degenerating axon profiles with very thin myelin and other axons with folded or split myelin were noted (Table 2).

### 5.3. Teres minor (n = 3) and Pax (n = 2)

Teres minor and Pax nerves displayed the most marked changes (Figure 2a–d). Myelinated fibre size, myelin thickness and density were obviously reduced. The perineurium was very thick and regions of the endoneurium appeared pale, oedematous, and had a disrupted architecture and areas devoid of cells (Figure 2a,d). In contrast, other regions appeared well-stained and housed numerous groups of unmyelinated axons surrounded by very small diameter myelinated axons (Figure 2a,c). Unlike other nerves described in this study, they contained Renaut bodies, i.e., distinctive endoneurial structures comprised of circular swirls of connective tissue that, in many cases, house circumferentially oriented cells with thin “spidery” cytoplasmic extensions (Figure 2a,b). The endoneurial blood vessels were prominent and had a thickened vessel wall (Figure 2c).

## 6. Recipient Nerves

### 6.1. PIN (n = 4) and AIN (n = 2)

The PIN and AIN fascicles were dissected out from their parent nerves. They comprised two or more irregularly shaped fascicles and both longitudinal and transverse axon profiles appeared in the same sections (Figure 3a–c). In all specimens, many axons were surrounded by thickened or folded myelin. Degenerating axons were also observed. They were surrounded by a “normal” or fragmented myelin sheath (Figure 3b) or were absent (Figure 3e). Degenerating axons are identified by their detachment from the myelin and their vacuolated appearance, or, in some cases, by fragmentation of the fibre. A pale amorphous material, possibly uncompacted myelin was observed surrounding typical dark staining myelin (Figure 3c,d). Uncompacted myelin occurs when the lipid lamellae of the myelin component separate. This changes the staining characteristic of the myelin. This feature is best observed at the electron microscope level (not done). In four out of six nerves, there was a population of large de-myelinated axons (Figure 3e). Small thinly myelinated axons and unmyelinated axons were observed in all specimens (Figure 3b). In one specimen, there was also a marked reduction of large myelinated axon density and only a sparse population of small, thinly myelinated and unmyelinated axons (Table 2).

### 6.2. Triceps (n = 3)

Each of the triceps’ nerves differed in morphologic appearance. One was relatively normal, except for the many mononuclear cells in the surrounding blood vessels (Table 2). The second had myelin abnormalities in most of the fibres, small thinly myelinated axons, many groups of small unmyelinated axons, large demyelinated axons, and blood vessels filled with mononuclear cells (see Figure 4). The third showed myelin abnormalities (as described above) but also a markedly reduced large myelinated axon density and both transverse and longitudinally oriented axon profiles (Figure 4a–d).

## 7. Discussion

Multiple nerve transfer surgery is increasingly used to restore motor function of the upper limbs following cervical SCI, and is becoming the method of choice in suitable candidates [2]. In this study, we obtained multiple nerve biopsies from SCI patients at the time of nerve transfer surgery. Although it is generally assumed that peripheral nerves technically outside the SCI zone are relatively normal, we identified abnormalities in ~80% of nerves. These abnormalities fell into three main categories: firstly, widespread myelin abnormalities; secondly, a reduction of large myelinated axon density; and thirdly, inflammation.

With respect to the donor nerves examined in this study, the most unexpected finding was that of marked abnormalities in all specimens of the posterior axillary nerve and the nerve to the teres minor muscle. These donor nerves branch from the axillary nerve. There was an obvious reduction in large myelinated axon density, a population of very thinly myelinated axons, and many foci of small unmyelinated axons. All nerves had a thickened perineurium and an oedematous endoneurium. Inflammatory cells (monocytes) were noted in the perineural and endoneurial blood vessels, and the endoneurial blood vessel wall was thickened. Specifically, Renaut bodies were identified in the sub-perineurium and within the endoneurium of all specimens. Similar observations have been made in one case study of superficial radial nerves known to have been subjected to chronic compression [16].

Renaut bodies are “cylindrical, hyaline appearing, loosely-textured, whorled, cell-sparse structures” composed of collagen, elastin precursors and spidery fibroblasts. They measure 20–130 μm in diameter, are hundreds of μm long and may have a perineural origin [17,18,19]. Their significance is unknown, but they are present in >50% of nerves at risk of subclinical entrapment with or without fibre pathology [20,21]. The axillary nerve is at risk of compression. It courses through the quadrilateral space with the posterior circumflex humeral artery, where it can be subjected to friction and compression by the surrounding tendons. This irritation often gives rise to a ganglionic enlargement of the nerve with pain (known as quadrilateral space syndrome), or without pain, and may result in teres minor muscle atrophy [22,23,24]. In studies of twenty cadavers, degenerated nerves were identified in 75% of shoulder joints, and 60% of these occurred in nerve branches “likely to be” from the axillary nerve. [25,26]. Our observations suggest that, in these three patients, the Pax and TM nerves were subjected to chronic compression that is likely to have occurred prior to the nerve transfer procedure. Since all the Pax and TM nerves examined were affected, there may be predisposing factors in this group of patients. Notably, two additional donor nerves, one supinator and one ECRB, also had a markedly reduced axon density and clusters of regenerating axons but no Renaut bodies. Consistent observation of numerous clusters of small unmyelinated axons and thinly myelinated axons in specimens with reduced large axon density could be attributed to axonal sprouting, regeneration and remyelination in response to ongoing injury [16].

Recipient nerves also exhibited abnormalities. Unlike the donor nerves, there was good axon density but markedly abnormal myelination. This included myelin sheaths that were frequently folded, thickened, split or degenerated, and, in some cases, appeared to have undergone de-compaction. Some axons were compressed or obliterated. Large unmyelinated axons as well as the occasional degenerating axon were also observed. For the most part, these observations would be classified as reversible, Sunderland grade 1 or Neurapraxia lesions (i.e., “segmental myelin damage with axons intact”) that cause a reversible conduction block that is restored after remyelination occurs” [27]. Our observations in human peripheral nerves are similar to those observed in specimens of caudal peripheral nerves sampled in a rat SCI model at three months post injury. In this rat model, axon numbers were maintained, although many had a detached (split) myelin sheath. Some changes corresponded to the Schmidt–Lantermann incisures near the nodes of Ranvier. No axon degeneration was observed. However, a decreased compound muscle action potential (CMAP) amplitude was recorded, and this was inversely related to the severity of the SCI. The investigators attributed the axonal observation to nerve compression [28].

Interestingly, similar myelination abnormalities have also been demonstrated in human sural nerve biopsies from patients diagnosed with demyelinating peripheral neuropathies, such as Charcot–Marie–Tooth, and in experimental models of hereditary motor and sensory neuropathy that have altered Schwann cell–axon interaction. In these heterogeneous neuropathic syndromes, abnormal myelin proteins or anti-myelin antibodies lead to cell adhesion abnormalities, impaired maintenance of the myelin sheath, demyelination and ultimately axon death [29,30,31,32]. The myelin changes are accompanied by decreased nerve conduction velocity, reduced CMAP amplitude, and impaired axonal transport [33,34].

There are no predisposing genetic factors in this group of SCI patients that could account for the breakdown of myelin integrity, so factors relating to the injury must be considered. One such factor may be the abrupt cessation of upper motor neuron (UMN) trans-synaptic input and peripheral nerve conduction. Axons and Schwann cells rely reciprocally on each other for the appropriate signals in order to maintain integrity. Studies in frogs show that nerve conduction of electrical impulses evokes increased calcium levels in the Schwann cells, at the nodes of Ranvier and at the neuromuscular junctions. This calcium mediated activity-dependent communication between axons and Schwann cells has been suggested to play a role in the maintenance of the neuromuscular synapse, Schwann cell and myelin integrity [35,36]. Furthermore, cessation of axon conduction results in reduced axonal transport of proteins that are critical for myelin maintenance and other proteins that are necessary for neuromuscular junction (NMJ) transmission [37,38]. This reduced transport may also contribute towards myelin and NMJ abnormalities. Axonal stimulation and functional electrical stimulation (FES) reintroduce axonal conduction and have a potential role in maintaining the integrity of peripheral nerves and NMJ. A six-week protocol of peripheral nerve stimulation improved nerve excitability parameters in SCI patients [39].

Schwann cell and myelin abnormalities in the peripheral nervous system (PNS) probably contribute to the electrophysiological abnormalities recorded in peripheral nerves and target muscles. These include absent or reduced nerve conduction and CMAP amplitudes, muscle fibrillations, aberrant neuromuscular junction transmission, and reduced nerve conduction velocity [12,40,41]. Abnormalities in axonal excitability, leading to higher stimulation thresholds, have been reported using more sensitive electrophysiological methods. These abnormalities are classically attributed to axon loss, although there is little information regarding ventral horn motor neuron numbers in chronic SCI cases [40,41,42]. In at least one human study, L3 ventral horn cell-profile numbers in four chronic complete C5/6 SCI cadavers were similar to that in controls [43]. In a mouse study, an unbiased stereology technique was used to show that ventral horn motor neuron numbers were not decreased after spinal cord transection [44]. Others attribute abnormalities to nerve compression or traction [42,45,46] or to changes in LMN axonal structure, ion channel function, decentralisation and inactivity [12,41,46].

An inflammatory response was noted in one-half of the donor nerves and one-half of the recipient nerves, based on the observation of perineural and endoneurial vessels filled with mononuclear blood cells. Elevated levels of the inflammatory cytokines, interleukin-2 and Tumor necrosis factor have been described in the peripheral circulation of a sub-population of SCI patients [47]. The significance of these cytokines in the PNS is unknown, but they contribute to demyelination in some central nervous system diseases. Inflammation in the PNS of SCI patients may either contribute to, or be a response to, the abnormalities observed and merits further investigation.

## 8. Conclusions

In SCI, the ventral horn motor neurons and axons below the injury are at risk of degeneration even though they are outside the injury zone. We investigated the morphology of peripheral nerve biopsies from cervical SCI patients undergoing multiple nerve transfers and have described abnormalities both above and below the SCI that have not been previously described in SCI. We propose that some donor nerves are subjected to ongoing compression post-SCI but show a robust regenerative response. Some recipient nerves undergo myelin changes that may eventually lead to demyelination and neuron death, but these changes may also be reversible. This axon population also shows regenerative effort. Maintenance and support of the PNS is essential if treatments to restore voluntary movement after SCI are to be successful. An understanding of the impact of SCI on the morphology of PNS complements and sheds light on the electrophysiological observations and may also guide interventions that slow “degenerative” processes and restore volitional control. The PNS abnormalities described here have not been previously described in SCI.

## Figures and Tables

**Figure 1 brainsci-06-00042-f001:**
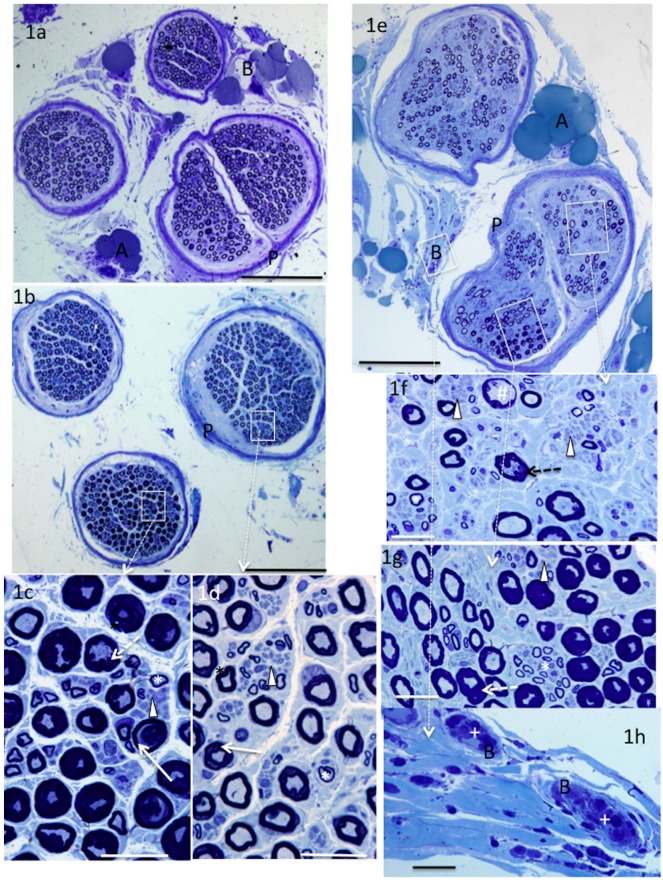
Methylene blue-stained sections from supinator nerves. Supinator nerves showed morphologic abnormalities in two patients. Normal nerves showed fascicles containing large myelinated axons of relatively uniform size, myelin thickness and orientation (**a**); an abnormal nerve (**b**) with one fascicle mostly occupied by axons surrounded by markedly thickened; split or folded myelin (**c**) and one fascicle less affected (**d**). A severely affected nerve has a reduced large myelinated axon density (**e**,**f**) in addition to thickened, folded, distended myelin (**f**,**g**). Note the groups of small unmyelinated axons (arrowheads), small thinly myelinated axons (*), thinned myelin (#), intravascular mononuclear cells (+), blood vessels (B), adipose cells (A), perineurium (P), folded myelin (dashed arrows) split myelin (arrows). **White** boxes highlight regions of higher magnification bar = 200 μm (**a**,**b**,**e**), 20 μm (**c**,**d**,**f**–**h**).

**Figure 2 brainsci-06-00042-f002:**
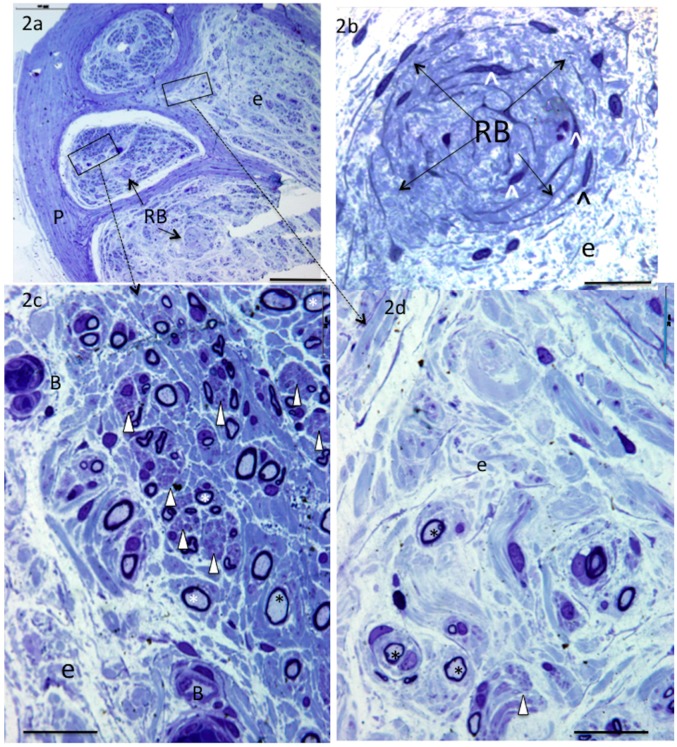
Toluidine blue-stained sections from nerves to teres minor. Nerves to teres minor and the posterior branch of axillary showed evidence of compression injury. Note thickened perineurium (**a**); reduced large myelinated axon density (**a**,**c**,**d**); Renaut bodies (**a**,**b**); oedematous, disrupted endoneurium (**a**,**b**,**d**); thick-walled blood vessels (**c**) regions of good (**c**) or sparse (**d**) best density of small thinly myelinated (*) and small unmyelinated axons (arrowheads) (in **c**), blood vessels (B), endoneurium (**e**), perineurium (P), Renaut bodies (RB), and spidery fibroblasts (^). Bar = 200 μm (**a**), 20 μm (**b**–**d**).

**Figure 3 brainsci-06-00042-f003:**
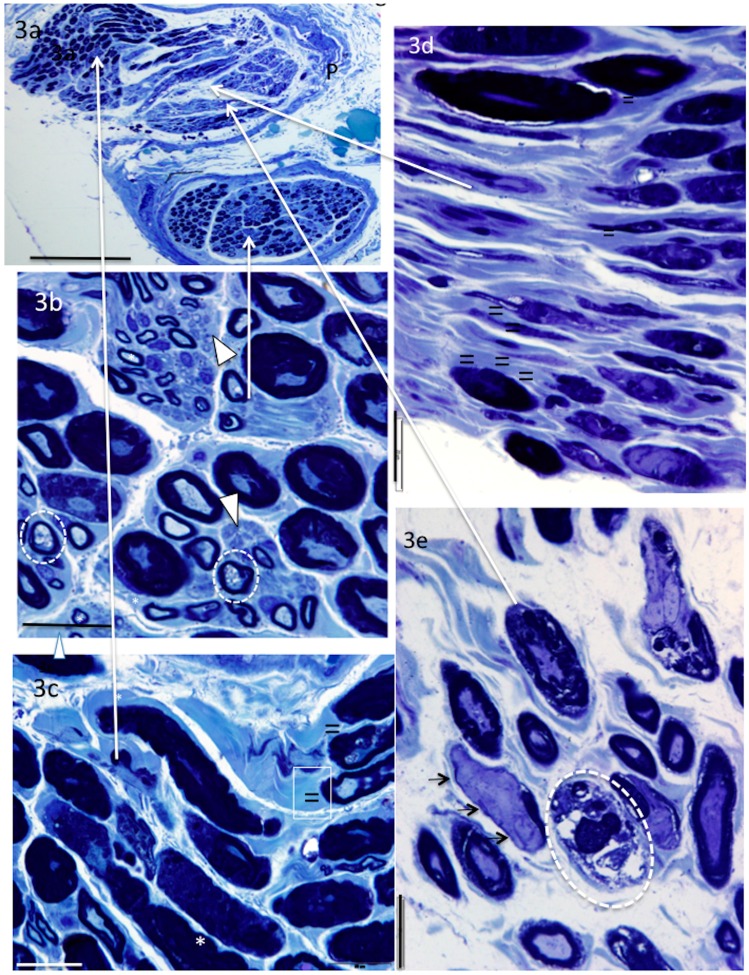
Methylene- (**a**–**c**) and toluidine (**d**–**e**) blue-stained sections from AIN. In addition to thickened, split and folded myelin, AIN and PIN nerves showed both transverse (**b**,**e**) and longitudinal orientation of axons (**c**,**d**). Abnormally pale-stained myelin (possibly uncompacted) can be seen surrounding the typically dark staining myelin (**c**,**d**). Note the large demyelinated axons, clearly lacking the myelin component (**e**) and degenerating vacuolated axons (**b**) or myelin without an axon (**e**). Small myelinated (*) and unmyelinated axons (arrowheads) were also observed. Demyelinated axons (small arrows), abnormally pale stained myelin abutting normal myelin (=), degenerating axons (enclosed in **white** circles). Perineurium (P), long arrows show region of high oower micrographs bar = 200 μm (**a**), 20 μm (**b**–**e**).

**Figure 4 brainsci-06-00042-f004:**
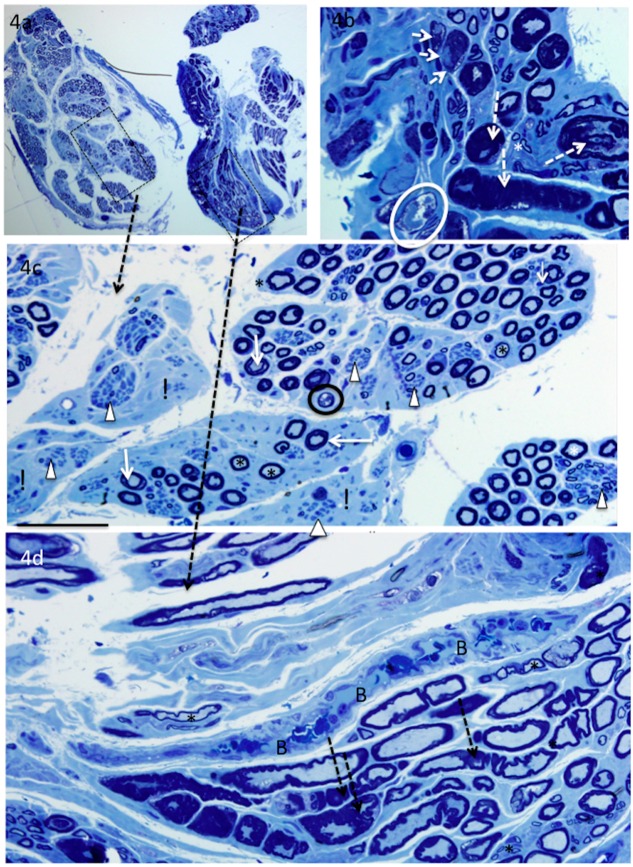
Methylene-blue stained sections from the most affected triceps nerve showing longitudinally and transversely oriented axons (**a**) and a wide range of fibre morphology. In one fascicle (**a**,**c**), there is a marked reduction in the density of large myelinated axons and some regions that are devoid of myelinated axons (!). There are numerous groups of small unmyelinated axons (arrowheads), and many small thinly myelinated axons (asterix). In the other fascicle (**a**,**b**,**d**), the changes consist mainly of thickened, folded or split myelin, and intravascular mononuclear cells (**d**). Degenerating fibres are observed in both fascicles. Typically, the axon detaches that from the myelin and may become vacuolated before being phagocytosed. Note the thickened and/or folded myelin (dashed arrows), split myelin (**white** arrows), abnormally pale stained myelin (=), large demyelinated axons (short **white** arrows), degenerating axons (circled) and mononuclear cell filled vessel (B). Bar = 200 μm (**a**), 20 μm (**b**), and 50 μm (**c**–**d**).

**Table 1 brainsci-06-00042-t001:** Patient demographics.

Patient Number	Patient Details
3	Male age 36 years
	TPI 46 weeks
	C5 ASIA B
6	Female Age 33
	TPI 46 weeks
	C5 ASIA A (C6 zone of partial preservation)
7	Male age 23 years
	TPI 46 weeks
	C5 ASIA A (C5 zone of partial preservation)

**Table 2 brainsci-06-00042-t002:** Qualitative assessment of nerve specimens.

-	Nerve	Axon Density	% Abnormal Myelin	Myelin	UM	-	Inflammation	-
3L	PIN	N	36%		+		I	T
3R	PIN	N<	89%		+			TL
6R	PIN	N<	100%	LDA D	+			TL
7R	PIN	N<	58%	LDA D	+		I	T
3R	AIN	N	27%	LDA	+			TL
6R	AIN	N<	58%	LDA	+			TL
3R	TRI	N	-		+++		I I I	T
6R	TRI	N<	43%	LDA D	+++		I	TL
7R	TRI	N<	69%	LDA D	+++		I	T
3L	**SUP**	N	-		+			T
3R	**SUP**	N	-		+			T
6R	**SUP**	N	30%	D	+		I	T
7R	**SUP**	N<	-	D	+		I	T
3R	**BRA**	N	-		+			T
6R	**ECRB**	N<	-	D	++		I	T
3R	**TM**	N<	-		++	RB		T
6R	**TM**	N<	-		+++	RB		T
7R	**TM**	NN<	-		++	RB	I	TL
6R	**PAX**	N<	-		+++	RB	I	TL
7R	**PAX**	N<	-		+	RB	I	T

This table shows the details of all nerves assessed. Each line represents data from one nerve. Recipient nerves are listed first: (PIN) posterior interosseous, (AIN) anterior interosseous and (TRI) triceps. Donor nerves (bolded text) comprise (SUP) supinator, (BRA) brachialis, (ECRB) extensor carpi radialis brevis, (TM) teres minor and (Pax) Posterior axillary nerve. The status of nerves in column (C) 3 is depicted as follows: colours denote a group of features with respect to axon density and extent of myelin changes as described below, with N representing relative normal specimens and N
N
N
N increasing severity of changes in specimens, A(<) denotes a reduced axon density. The other colours represent other individual features observed as described below: N = specimens with a good axon density/size/myelin thickness and few if any myelin abnormalities, N< = a notably reduced axon density/size/myelin thickness and few if any myelin abnormalities, N = good or N< reduced axon density, and up to 50% axons with thickened, folded, split myelin, N< = reduced large axon density and over 50% axons with thickened, folded, split myelin, N< = markedly reduced axon density and axon size, thickened perineurium, oedematous endoneurium and the presence of RB (Renaut bodies) **C7**. The % of axons associated with thickened/split/folded myelin is shown in **C4**. Other myelin abnormalities include LDA (large demyelinated axons) and D (degenerating axons) in **C5**. Semi-quantitative assessment of unmyelinated axon numbers (+ to +++) in **C6**. Observation of “inflammation” I (intravascular and extravasated mononuclear blood cells) **C8**. Axon orientation is (**T** = transverse), (**L** = longitudinal) **C9**.

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
