# Peer review of "Morphology of Donor and Recipient Nerves Utilised in Nerve Transfers to Restore Upper Limb Function in Cervical Spinal Cord Injury"

_brainsci, 2016, doi:10.3390/brainsci6040042_

Round 1

Reviewer 1 Report

This paper is a well done study with only a couple of insignificant typos.

Author Response

Thanks for reviewing the manuscript

 we hve reworked some of the less clear area and are not ready to resubmit

Regards Aurora

Reviewer 2 Report

This paper sought to document morphologic features of donor and recipient nerves being utilized in transfers for recovery of hand function in SCI patients. This is an important topic given the significance of regaining hand function for tetraplegic individuals (increased autonomy in ADLs, etc.) and has the potential to enhance our understanding of long-term clinical deficits present in SCI patients and inform decisions as to potential treatment options.

The quality of the manuscript is generally good and for the most part is well-written. The introduction is clear and outlines the rationale for the experiment adequately. The methods are adequately described and generally allows for interpretation of the figures. This is a qualitative, descriptive study, however there are a few instances where reference is made to quantitative aspects of axons. For example, one of the morphological characteristics indicated in Table 1 is whether < 30% of axons were filled with folded or split myelin versus > 50% of axons with the same characteristics. This suggests that some type of quantitative analyses were performed which should thus be specified in the methods section. Which sections, how many fields per sections, etc. were evaluated? Presenting an evaluation of axon: myelin ratios could be useful in quantifying some of the morphological changes.

The greatest weakness of the manuscript is in regard to the presentation of the results, including the figures. In general, the text and figures should be revised to better support each other. Because this is a descriptive paper, there needs to be greater detail describing the key morphological features that are indicated in the figures. Similarly, the figures should be revised to better highlight certain of the morphological features being described in the text. Specific suggestions include the following:

a) Clarify what is meant by “folded” and “split” myelin

b) Adjust figures and figure legends so that panels are discussed in sequence (all figures), and make sure that the legends describe each of the panels (e.g., figure 3 legend fails to include panel e)

c) Reference Table 1 when describing results that are not indicated in a figure, but are included in the table (e.g., ECRB nerve morphology)

d) In figure 2, it would be helpful to include a normal teres minor or pax nerve to highlight the myelin changes demonstrated in the figures shown

Additional specific comments on figures:

Figure 1. Please reorganize figure so that results are described in the order the figure panels are presented. Figure 1f appears to have a misplaced asterisk. Figure 1b has a “B” that should be on a blood vessel, but does not appear to be in quite the right place; what is the difference between red and white arrowheads? The “e” indicating epineurium does not aid the figure and is somewhat distracting in some instances as it cannot readily be appreciated with this type of stain. Suggest indicating it only when demonstrating that its appearance is edematous or otherwise exaggerated (i.e., keep in panel 1c but remove from 1a and 1b – or add it to the higher magnification panels)

Figure 2. Are panels b-d higher magnification of 2a? If yes, please indicate where they are taken from – may be useful to highlight in 2a the regions of disrupted endoneurium (highlighted in 2d) and the regions of more normal appearing endoneurium (highlighted in 2c); one of the asterisks in panel 2c seems to be misplaced

Figure 3. Results text and figure legends have errors in which panels are indicating demyelinated axons and abnormally pale stained myelin; the figure legend indicates both are present in panel d but the text refers to a population of large demyelinated axons in panel 3e; this needs to be clarified and indicated in the appropriate panel as it is not readily appreciated. It would be helpful to readers to indicate in the figure panels which are considered large demyelinated axons, which are axons surrounded by abnormally pale myelin sheaths, which are small myelinated axons and which are groups of unmyelinated axons. Currently, the arrowheads seem to indicate regions of the nerve where each of these types of morphologies are present, thus are not helpful to the reader in distinguishing these; there appear to be formatting issues in the pdf of the article with regard to the results text and the figure legend for Figure 3. It is not clear where the figure legend begins and ends. Panel 3d is not discussed in the results text. The # sign is supposed to specify large demyelinated axons and is present in panels 3b, 3c, and 3e; however, the legend specifies that this morphology is indicated in panel 3d

Figure 4. The text describing the figure can be much expanded and more detailed, given that this is the purpose of the paper. For example, what criteria were used to classify degenerating axons? This is a morphological finding first described and shown in this figure yet is not mentioned in the text of the results section

Table 1.

General comments: This table is somewhat difficult to read and interpret. As this data represents the major focus of the manuscript, it should be reworked. Perhaps the information could be separated into several smaller tables, or put into separate rows (e.g., axon density is currently combined with various measures of myelin, axon size, and findings related to connective tissue coverings). Advise separating these various indices and describing each individually. In the discussion, it is mentioned that the morphological abnormalities described fell into 3 main categories; perhaps the data contained in Table 1 could be stratified into these 3 categories to improve readability.

Specific comments:

a) The presence of asterisks associated with “N” is distracting and it is unclear why it is necessary to include them; suggest removing them

b) In the legend, there appears to be some information missing in the PDF – under “Additional features. Renaut bodies, there parentheses with no text included within them.

Author Response

Thankyou for  reviewing our manuscript and the very helpful comments.

We have reviewed much of the results section and the table for clarity and increased information

Regards Aurora

Reviewer 3 Report

This is an interesting report on the donor and recipients for SCI nerve transfers - not surprisingly there were changes to the segments caudal to the zone of injury - but this is reassuring they appear similar to a grade I injury. I think this paper adds to the literature and commend the authors on their work. I think the authors could clarify the table - the current presentation while informative is not presented clearly. 

Author Response

Thank you for reviewing our manuscript.

The table has been split into 2 parts and reworked for clarity

Regards Aurora